# Comparison of Precursor Preparation Routes on Final Density of Y_3_Fe_5_O_12_ Garnets Prepared via Reactive Sintering

**DOI:** 10.3390/ma14237316

**Published:** 2021-11-29

**Authors:** Kamil Wojciechowski, Radosław Lach, Magdalena Stan, Łukasz Łańcucki, Marta Gajewska, Dariusz Zientara

**Affiliations:** Faculty of Materials Science and Ceramics, AGH University of Science and Technology, 30-059 Kraków, Poland; kamilwoj@agh.edu.pl (K.W.); mstan@agh.edu.pl (M.S.); lancucki@agh.edu.pl (Ł.Ł.); marta.gajewska@agh.edu.pl (M.G.); zientara@agh.edu.pl (D.Z.)

**Keywords:** citrate precursor method, yttrium iron garnet (YIG), reaction sintering

## Abstract

Yttrium iron garnet was obtained using four methods of synthesis. A modified citrate method and a modified citrate method with YIG (yttrium iron garnet, Y_3_Fe_5_O_12_) nucleation were used. In two subsequent methods, YIP (yttrium iron perovskite, YFeO_3_) and α-Fe_2_O_3_ obtained in the first case by the citrate method and in the second by precipitation of precursors with an ammonia solution were used as the input precursors for reaction sintering. Differential scanning calorimetry (DSC) measurements of the output powders obtained by all methods allowed to identify the effects observed during the temperature increase. Dilatometric measurements allowed to determine the changes in linear dimensions at individual stages of reaction sintering. In the case of materials obtained by the citrate method, two effects occur with the increasing temperature, the first of which corresponds to the reaction of the formation of yttrium iron perovskite (YIP), and the second is responsible for the reaction of the garnet (YIG) formation. However, in the case of heat treatment of the mixture of YIP and α-Fe_2_O_3_, we observe only the effect responsible for the solid state reaction leading to the formation of yttrium iron garnet. The obtained materials were reaction sintered at temperatures of 1300 and 1400 °C. Only in the case of material obtained from a mixture of perovskite and iron(III) oxide obtained by ammonia precipitation at temperature of 1400 °C were densities achieved higher than 98% of the theoretical density. The use of Hot Isostatic Pressing (HIP) in the case of this material allowed to eliminate the remaining porosity and to obtain full density.

## 1. Introduction

Yttrium iron garnet (Y_3_Fe_5_O_12_) is one of the more important materials with ferromagnetic properties. It can be the base material for applications as an amplifier in microwave devices, circulator, insulator, phase shifter, etc. The numerous applications of YIG are due to its improved electromagnetic properties, including low dielectric loss, narrowing of the resonance linewidth in the microwave area, and controlled magnetization saturation [1,2,3,4,5,6,7,8].

Numerous literature reports concern the production of monocrystalline and polycrystalline YIG in the form of thin films [9,10,11,12]. On the other hand, very interesting properties could be exhibited by dense sinters of polycrystalline Y_3_Fe_5_O_12_. The traditional method of obtaining YIG from a mixture of yttrium oxide and iron oxide is based on a solid state reaction. Heating such a mixture at temperature of about 1300 °C leads to the formation of yttrium iron garnet. In this method, it is very difficult to achieve an ideal homogenization of the system. In addition, as a result of roasting at high temperatures, there is significant grain growth and the formation of mechanically durable agglomerates [13,14].

To obtain the desired morphology of the powder and dense sinters of YIG, many different techniques of obtaining the initial powders were used, such as co-precipitation [15,16,17], sol-gel method [5,18,19], synthesis from microemulsions [20], a method using organic precursors [5,21,22,23], a hydrothermal method [24], and mechanochemical technique. Wet chemical methods allow for a better homogenization of the initial mixture and for obtaining sinters with a smaller grain size.

First attempts to obtain dense YIG sinters using the microwave sintering method were made in 1970s. However, as a result of absorbing the microwave energy, there is a very rapid grain growth [25,26].

Attempts were also made to improve the density of the obtained sinters using 5, 15, and 20 wt.% addition of excess Fe_2_O_3_. However, this leads to YIP (YFeO_3_) inclusions in the Y_3_Fe_5_O_12_ matrix, and it changes the properties of the material [27,28,29].

Another method of improving the microstructure and density of the obtained sinters was based on reactive sintering of a precursor mixture of YFeO_3_ and Fe_2_O_3_. Using this method allows one to obtain 97.8% of the theoretical density at temperature of 1425 °C. The holding time at this temperature is 16 h. Regrettably, such elevated temperature leads to rapid grain growth and prevents the complete elimination of internal porosity [30].

Thus, literature data still lack easy and cost-effective methods of synthesis of dense yttrium-iron garnets. Most of the methods described so far require long-term grinding of the precursor powders [30,31,32,33]. Moreover, in most cases, temperatures of 1400 °C and even 1450 °C are used for a very long hold time (up to several days), thus the final materials obtained in such way have final concentration of 96% of the relative density, which is insufficient. In this manuscript four different methods of obtaining Y_3_Fe_5_O_12_ were presented, using as precursors powders obtained by the citrate method; the citrate method using heterogeneous nucleation of the Y_3_Fe_5_O_12_ phase; mixture of YFeO_3_ and α-Fe_2_O_3_ where the starting materials were obtained by the citrate method; and the ammonia precipitation method. At present, no data describing a comparison of such methods have been presented in the literature. Additionally, the energetic effects occurring during the reactive sintering in the case of the presented methods were presented and described in detail.

## 2. Experimental

Method A—yttrium iron garnet obtained by a modified citrate method.

Method B—yttrium iron garnet obtained by a modified citrate method with a 10 wt.% addition of YIG (Y_3_Fe_5_O_12_) as embryos.

Method C—yttrium iron garnet obtained from a mixture of YIP (YFeO_3_) and α-Fe_2_O_3_ obtained by the citrate method.

Method D—yttrium iron garnet obtained from the mixture of YIP (YFeO_3_) and α-Fe_2_O_3_ obtained as a result of roasting of precursors precipitated with an ammonia solution.

### 2.1. The Citrate Method

The study used the solution of iron(III) nitrate obtained by dissolving Fe(NO_3_)_3_ · 9 H_2_O with analytical grade purity (Polskie Odczynniki Chemiczne S.A., Gliwice, Poland) and Y_2_O_3_ powder with 99.9% purity (Sigma Aldrich, Saint Louis, MO, USA).

The appropriate weighted amounts of yttrium oxide were subjected to hot dissolution process in concentrated nitric acid of analytical grade purity (Polskie Odczynniki Chemiczne S.A., Gliwice, Poland). Subsequently, an aqueous solution of iron(III) nitrate was introduced into the yttrium nitrate solution thus obtained in an amount corresponding to 3/5 molar ratio of Y^3+^/Fe^3+^ ions. Citric acid was then introduced into the obtained solution in such a quantity that the ratio of the sum of Y^3+^ and Fe^3+^ ions to the citric acid was 1:2. The 4% PVA solution in a 3:1 volume ratio was added to the mixture to prevent accidental pyrolysis during heating. The resulting powder mixture was dried at 120 °C for 24 h.

### 2.2. Precipitation with an Ammonia Solution

For the study, iron(III) nitrate nonahydrate with analytical grade purity (POCH, Gliwice, Poland) and commercial Y_2_O_3_ (Sigma Aldrich, Saint Louis, MO, USA) with 99.9% purity were used. The yttrium nitrate solution was obtained by dissolving an appropriate amount of yttrium oxide in nitric acid, while the solution of iron(III) nitrate was obtained by dissolving iron(III) nitrite nonahydrate in distilled water. Then the resulting aqueous solutions were mixed in the right proportions. The thus obtained solution of nitrates was introduced to vigorously stirred solution of ammonia. During the entire precipitation process, as well as after its completion, the pH of the suspension was maintained at 11, which ensured quantitative precipitation of the precursor. The resulting sludge was washed with distilled water until nitrate ions disappeared in the detection reaction. The resulting sludge was washed several times with isopropyl alcohol to remove water, then vacuum filtered on a Büchner funnel and dried at 120 °C for 24 h.

Method A

The precursor in which the molar ratio of Y^3+^/Fe^3+^ ions was 3/5, which corresponds to the stoichiometry of yttrium iron garnet was obtained by the citrate method. The obtained powder was roasted at 600 °C for 3 h to obtain the solution of Y_2_O_3_ and α-Fe_2_O_3_.

Method B

YIG (Y_3_Fe_5_O_12_) embryos in the quantity of 10 wt.% were introduced into the powder obtained by Method A. The use of YIG phase nucleation can lower the temperature of the solid state reaction between YIP (YFeO_3_) and α-Fe_2_O_3_.

Method C

Precursors of perovskite YIP (YFeO_3_) and iron oxide were obtained separately by the citrate method. Then, to obtain YIP and α-Fe_2_O_3_, the precursors were roasted at temperature of 800 °C and 400 °C, respectively.

Method D

Precursors of perovskite YIP YFeO_3_) and iron oxide were obtained separately by ammonia precipitation. Then, to obtain YIP and α-Fe_2_O_3_, the precursors were roasted at temperature of 800 °C and 400 °C, respectively.

The powders obtained by all methods (A, B, C, D) were sifted on a 90 µm sieve. All powder mixtures (B, C, D) were homogenized in a roller mixer in isopropanol for 24 h.

From the obtained powders, uniaxial cylindrical compacts with a diameter of 18 mm and a height of 2.5–3.0 mm at a pressure of 50 MPa were pressed and then isostatically ironed at a pressure of 250 MPa. Then, the compacts were sintered in air atmosphere at 1300 and 1400 °C with the rate of temperature increase 10 °C/min and 2 h soaking time.

Powders were characterized by the specific surface area measurements using nitrogen adsorption (Quantachrome, Nova 1200, Anton Paar GmbH, Ostfildern, Germany). The DSC data (Netzsch STA 449 F3 Jupiter, Netzsch Instrumenty Sp. z o.o., Kraków, Poland) were used to follow the phenomena occurring during the heat treatment of the powder sample. Measurements were made in the air atmosphere. The rate of temperature increase was set at 40 °C/min. Such a heating path was chosen to identify heat transfer effects attributed to the YIG formation that were problematic to observe at slower heating rates. Phase composition of the samples was determined by the X-ray diffraction (CuKα radiation, equipment Empyrean, Malvern Panalytical, Malvern, UK). The information on the powder densification process via compact shrinkage was obtained using a dilatometer (DIL 402C, Netzsch Instrumenty Sp. z o.o., Kraków, Poland). Hydrostatic weighing allowed us to determine the apparent density of the sintered samples. Microphotographs of the obtained samples were taken using a scanning electron microscope (SEM) (FEI Nova Nano SEM 200 Hillsboro, OR, USA). The quantitative description of sintered garnets microstructure was calculated according to methods described by Mendelson et al. [27]. To eliminate the remaining pores and to obtain full density, the selected materials were subjected to hot-isostatic pressing (HIP) (EPSI HIP 400-77^*^150 GM, EPSI, Temse, Belgium). The samples were placed in a corundum crucible in a bed of hexagonal boron nitride powder and the HIP process was performed at 1400 °C for 2 h under 200 MPa and temperature rate of 10 °C/min.

## 3. Results and Discussion

The experimental data consisting of specific surface area are presented in Table 1. The largest grain size disproportion between the perovskite and iron oxide grains was observed in the case of the powder obtained by the method D. This may be the result of large perovskite grains that were surrounded by fine iron oxide grains. This should increase the number of contacts between both phases, which should facilitate the reaction between the substrates. This is in agreement with data presented by Young et. al. [30]. However, in the case of Methods A and B, we cannot conclude what is the difference in the grain size of perovskite and iron oxide because YFeO_3_ and α-Fe_2_O_3_ crystallize only at temperatures above 700 °C.

DSC measurements (Figure 1A–D) of the initial powders obtained by all methods allowed to identify the effects observed during the temperature increase. Figure 1A,B show two effects each, first at temperature of 778.1 °C (Figure 1A) and 762.9 °C (Figure 1B), respectively, which corresponds to the reaction throughout yttrium iron perovskite (YFeO_3_) is created. Whereas, in the case of all methods, we observe an effect above 800 °C, for which a reaction in the solid state between iron oxide and yttrium iron perovskite is responsible, in which the yttrium iron garnet (Y_3_Fe_5_O_12_) is created. The lowest temperature of Y_3_Fe_5_O_12_ creation can be observed for Method B, which is most likely due to the use of YIG embryos. For all synthesized samples, we observe a weight loss of up to 3% at temperature till about 600 °C. This occurrence may correspond to the desorption of occluded water and other gases adsorbed on the surface of the powders.

For the powders obtained by Method A and Method B, we can observe additional decrease in mass at the temperature of yttrium iron perovskite (YFeO_3_) formation (at temperature of 778.1 °C (Figure 1A) and 762.9 °C), which may be caused by formation of crystalline structure from the amorphous phase.

This effect was not observed for the powders synthesized by Method C and D as presented in Figure 1C,D). Diffraction measurements (XRD) were performed in order to confirm the reactions taking place with the temperature increase, the results of which correlate with the effects observed on the DSC curves. For the powders obtained with Methods A and B, two effects were observed. Thus, additionally XRD analysis of the precursor powder synthesized via Method A, calcined at 600 °C, was performed. The only phase observed in this case was the amorphous phase as presented in Figure 2A. Consequently, the powder was further heated up to a temperature of 800 °C and rapidly cooled to room temperature. For such treated sample the phases observed were yttrium iron perovskite (YFeO_3_) and α-Fe_2_O_3_ (Figure 2A). The powder was further heated to a temperature of 1100 °C. After such heat treatment only phase corresponding to yttrium-iron grenade was observed (Figure 2A). Similarly, in the case of materials obtained from a mixture of perovskite and α-Fe_2_O_3_. The precursor powder obtained by Method C was firstly calcined at 800 °C and further at 1100 °C, respectively. At the temperature of 800 °C a mixture of yttrium iron perovskite (YFeO_3_) and α-Fe_2_O_3_ was observed, while for samples further heated at the temperature of 1100 °C the only observed phase was Y_3_Fe_5_O_12_ (Figure 2B). The obtained results agree with the literature reports approving that in the case of the mixture of Y_2_O_3_ and Fe_2_O_3_ the first reaction step is formation of perovskite structure YFeO_3_, and a further increase in temperature leads to further incorporation of iron leading to the formation of yttrium-iron garnet structure [30,31,32,33].

An analysis of dilatometric curves (Figure 3A–D) concludes that in the case of the precursor of yttrium iron garnet obtained by Method A and Method B at temperature of 751.2 °C and 742.9 °C, respectively, which corresponds to the reaction of the YFeO_3_ formation, we observe a faster change in the linear dimension of the sample. The reaction leading to the formation of perovskite is associated with a change in volume, where the volume of the resulting product is smaller than the volume of substrates involved in the reaction.
(1)$$Y_{2}O_{3}+{\;\mathrm{Fe}}_{2}O_{3\;}\to 2{\;\mathrm{YFeO}}_{3}$$
(2)$$\frac{225.81}{5.01\;}+\frac{159.68}{5.24}>\frac{2\cdot 192.75}{5.47}\;\left[\frac{\frac{g}{\mathrm{mol}}}{\frac{g}{{\mathrm{cm}}^{3}}}=\frac{{\mathrm{cm}}^{3}}{\mathrm{mol}}\right]$$
(3)$$75.54>70.48\;\left[\frac{{\mathrm{cm}}^{3}}{\mathrm{mol}}\right]$$

Whereas, in all other cases, respectively, at temperature of 895.6 °C (Method A) 832.9 °C (Method B), 1020.6 °C (Method C), and 974.9 °C (Method D), we observe the increase of the sample volume, resulting from the reaction of Y_3_Fe_5_O_12_ formation:(4)$$3{\mathrm{YFeO}}_{3}+{\;\mathrm{Fe}}_{2}O_{3\;}\to Y_{3}{\mathrm{Fe}}_{5}O_{12}\;$$
(5)$$\frac{3\cdot 192.75}{5.47}+\frac{159.68}{5.24}<\frac{737.94}{5.11}\;\left[\frac{\frac{g}{\mathrm{mol}}}{\frac{g}{{\mathrm{cm}}^{3}}}=\frac{{\mathrm{cm}}^{3}}{\mathrm{mol}}\right]$$
(6)$$136.18<144.41\;\left[\frac{{\mathrm{cm}}^{3}}{\mathrm{mol}}\right]$$

In the case of solid state reaction sintering, the most advantageous is the situation where the reaction products have a larger volume than the substrates. This causes the reaction products to fill the pores during sintering, which leads to a complete densification of the material. This situation is observed in the cases of sintering mixtures of YIP (YFeO_3_) and α-Fe_2_O_3_. However, in the case of sintering of precursors of yttrium iron garnet obtained by the citrate method, the reaction of the perovskite (YFeO_3_) formation causes additional shrinkage of the material and then in the reaction of the garnet formation, the resulting product has a larger volume than the substrates. This can cause changes in the microstructure, which consequently do not allow the material to be fully compacted.

Table 2 presents the results of the relative density of samples sintered at temperature of 1300 °C and 1400 °C. In the case of materials obtained by the citrate method, no density above 90% of the theoretical density was obtained, and a temperature of 1400 °C allowed us to achieve a maximum density of 89% of the theoretical density of samples containing 10 wt.% of YIG embryos. However, in the case of the material obtained as a result of reaction sintering of yttrium iron perovskite (YFeO_3_) with iron oxide, the density of 93% and 98%, respectively, was obtained. The lack of open porosity was only displayed by the sample obtained by Method D. The SEM microphotographs presenting obtained sinters are shown in Figure 4. In the case of this sample, Hot Isostatic Pressing (HIP) was additionally used, resulting in a density of 99.99% of the theoretical density.

A comparison of the size of grains in sinters obtained at 1400 °C presented in Table 3, shows that for all materials, the average grain size was in the range of 4.0–5.5 µm. As a result of the additional pressure applied, which leads to an increase in grain size in the case of the sample (Method D) subjected to Hot Isostatic Pressing (HIP), it was possible to eliminate the remaining porosity. Comparison of obtained sintering results with the existing literature data reveals that similar temperatures were used in order to obtain a relative density values up and above 96%, but holding time described in manuscripts was even up to 16 h [30]. In our studies, in the case of Method D, it was possible to obtain sinters of almost 99% density at 1400 °C using holding time of only 2 h. Despite such a long sintering time at the maximum temperature, it was not possible to obtain material exhibiting full compaction [30,31,32,33]. Only the use of Hot Isostatic Pressing (HIP) made it possible, as presented in Figure 5.

## 4. Conclusions

Analysis of DSC curves and XRD diffractograms of precursor powders heated at certain temperatures allowed us to identify the temperature range of yttrium-iron perovskite formation reaction (YIP) occurring in the case of Method A and Method B. In the case of these methods, the perovskite (YFeO_3_) formation reaction was associated with a change in volume, where the volume of the resulting product was smaller than the volume of the substrates involved in the reaction. Then, in the reaction of the garnet formation, the resulting product had a larger volume than the substrates. In these cases, there were most likely changes in the microstructure, which enabled the achievement of the high density of the material.

The obtained powders were reaction sintered at temperature of 1300 and 1400 °C.

In the case of materials obtained by Method A and Method B, it was not possible to obtain a density of more than 90% of the theoretical density. However, in the case of the material obtained as a result of reaction sintering of yttrium iron perovskite (YFeO_3_) with iron oxide, densities of 93% and 98%, respectively, were obtained. The lack of open porosity was only displayed by the sample obtained by Method D. In the case of this sample, Hot Isostatic Pressing (HIP) was additionally used to obtain full density of the material.

The average grain size of the samples obtained by all methods was in the range of 4.0–5.5 µm.

The highest density, that reached almost 100% of the theoretical, was obtained in the method of the reactive sintering of the mixture of YFeO_3_ and α-Fe_2_O_3_ obtained by precipitation of precursors with ammonia. Moreover, the use of Hot Isostatic Pressing (HIP) to eliminate residual porosity after free sintering did not result in significant grain growth. Identification of the effects that occur with increasing temperature shows that the perovskite formation reaction (YFeO_3_) during sintering significantly reduces the final material compaction, which, as indicated in the literature, can be achieved at the level of 96% relative density only after using a long holding time in the elevated temperatures. In addition, the use of heterogeneous nucleation with the Y_3_Fe_5_O_12_ phase shows that the effects of perovskite formation and the perovskite and garnet reaction are in a very narrow range, which may have a significant impact on densification, but this requires further research and the use of nucleation in other methods.

## Figures and Tables

**Figure 1 materials-14-07316-f001:**
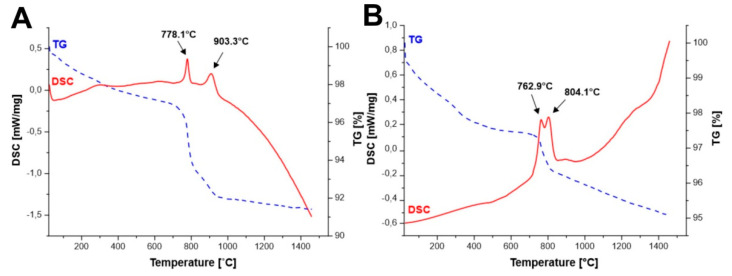
DSC curves of precursor powders obtained from: (**A**)—Method A; (**B**)—Method B; (**C**)—Method C; (**D**)—Method D.

**Figure 2 materials-14-07316-f002:**
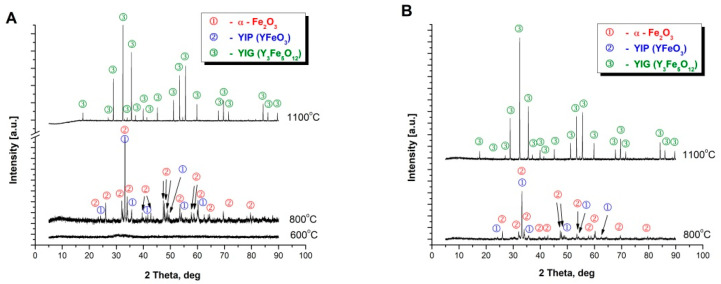
X-ray diffraction pattern of the powders: (**A**)—Method A; (**B**)—Method C.

**Figure 3 materials-14-07316-f003:**
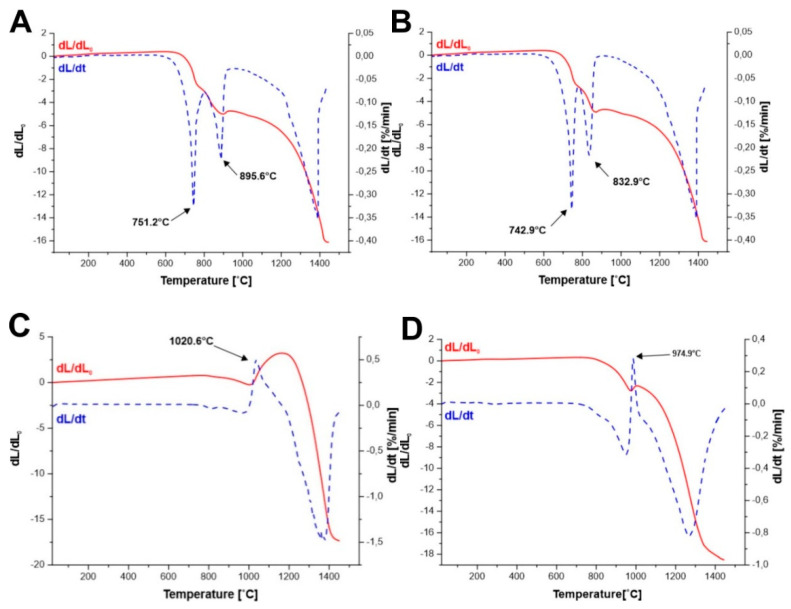
Dilatometric and derivative curves of precursor powders: (**A**)—Method A; (**B**)—Method B; (**C**)—Method C; (**D**)—Method D.

**Figure 4 materials-14-07316-f004:**
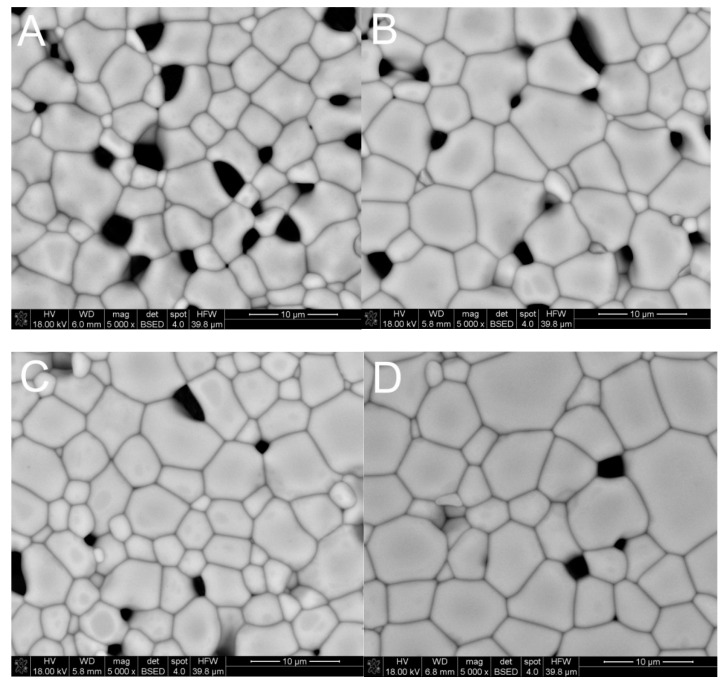
SEM micrographs of materials sintered at 1400 °C: (**A**)—Method A, (**B**)—Method B, (**C**)—Method C, (**D**)—Method D.

**Figure 5 materials-14-07316-f005:**
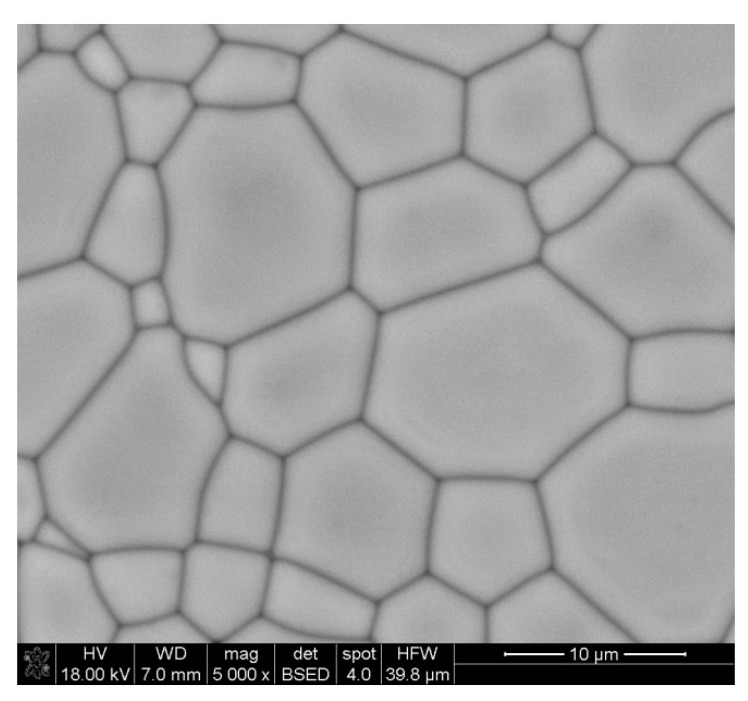
SEM micrographs of materials obtained by the Method D and subjected to Hot Isostatic Pressing (HIP).

**Table 1 materials-14-07316-t001:** Specific surface area and grain size.

	Method A	Method B	Method C		Method D	
S_w_ [m^2^/g]	19.1	18.4	YFeO_3_	Fe_2_O_3_	YFeO_3_	Fe_2_O_3_
			12.3	20.8	23.2	86.6
D_BET_ [nm]	64.2	63.5	89.0	55.1	47.3	13.7

**Table 2 materials-14-07316-t002:** Relative density (% theo) of the sintered samples.

Temperature/Method	Method A	Method B	Method C	Method D
1300 °C	78.34 ± 0.08	78.13 ± 0.03	90.73 ± 0.04	92.48 ± 0.03
1400 °C	85.29 ± 0.02	89.32 ± 0.02	93.03 ± 0.02	98.66 ± 0.02
1400 °C (HIP)	-	-	-	99.99 ± 0.01

±—confidence interval at 0.95 confidence level.

**Table 3 materials-14-07316-t003:** Average grain size of materials sintered at 1400 °C [34].

Method	Average Grain Size [µm]
A	5.33 ± 0.36
B	4.49 ± 0.29
C	4.09 ± 0.29
D	5.41 ± 0.36
D (HIP)	7.58 ± 0.48

±—confidence interval at 0.95 confidence level.

## Data Availability

Not applicable.

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
