# Peer review of "Comparison of Precursor Preparation Routes on Final Density of Y3Fe5O12 Garnets Prepared via Reactive Sintering"

_materials, 2021, doi:10.3390/ma14237316_

Round 1

Reviewer 1 Report

The paper looks at the effect of different precursor preparation routes on the final density of yttrium iron garnets made via reactive sintering. Four different methods are examined, three using the citrate method and one using ammonia precipitation. The paper makes an effort to compare the densities and grain sizes observed, with phase change mechanisms. For this it uses differential scanning calorimetry and dilatometry data. However, it is missing some crucial data, especially phase determination information. Without this experimental data, the results are inadequate to support the conclusions. Detailed comments are included below.

Major comments:

  1. The introduction section of the paper is short and I feel does not provide sufficient background, which makes it difficult to understand the hypothesis and goal of the work. The authors are suggested to include a general background on the desired electromagnetic properties, microstructures and more importantly how they relate to each other.
  2. Page1, Paragraph 3, Line 1: Instead of saying ‘the desired morphology’, it would help better understand the motivation for the experiments if the authors elucidated on exactly what the desired morphology and ideal microstructure is.
  3. Introduction: There is currently no connection between the ‘introduction’ section and the ‘methods’ section. It is not clearly established where the gap in the current knowledge and literature is. This again goes back to what the motivation is and what is needed to achieve the desired microstructure. It will be helpful if at this point, the authors clearly define a hypothesis which explains why they are they are trying the experimental plan presented in the paper, and how it might improve upon existing reaction sintering work, for example by Young et al. (reference number 26) cited in the paper.
  4. In section 2.1 and 2.2, it would be helpful to include a comment on the nature the intermediate powders formed from the citrate method versus those from precipitation with ammonia especially since the paper is studying the effect of the precursor routes.
  5. Page 3, Paragraph 8: In the ‘Results and discussion’ section, it is not clear how the authors identified the exact reactions and phases formed at the different temperatures simply based on the differential scanning calorimetry data. There are no references to prior work which has identified these phases and reactions. The data shown in the paper does not support the statements about YIP and YIG formation. The authors are suggested to include phase information from x-ray diffraction measurements at various temperatures like those done by Azis et al. (Azis, R. S., M. M. Syazwan, N. M. M. Shahrani, A. N. Hapishah, R. Nazlan, F. M. Idris, I. Ismail et al. "Influence of sintering temperature on the structural, electrical and microwave properties of yttrium iron garnet (YIG)." Journal of Materials Science: Materials in Electronics29, no. 10 (2018): 8390-8401). Another example is work done by Rodzia et al. (Rodziah, N., M. Hashim, I. R. Idza, I. Ismayadi, A. N. Hapishah, and M. A. Khamirul. "Dependence of developing magnetic hysteresis characteristics on stages of evolving microstructure in polycrystalline yttrium iron garnet." Applied surface science 258, no. 7 (2012): 2679-2685). Such an independent measurement is crucial to the hypothesis I think is being presented (since it is not explicitly defined) in the paper given that the authors are using novel processing routes not used before.
  6. Results and discussion: In figure 1, thermogravimetric data is shown along with calorimetric data, however no comments are made as to the observed weight changes and how it corresponds with the proposed reactions. The authors are suggested to include at least some comments on the significance or insignificance of these observations in the discussion as this information seems pertinent to the densities observed from the different precursor preparation routes.
  7. Results and discussion: The advantage of the different precursor routes is not exactly clear. If the goal is to achieve the highest possible density while keeping the grain growth under control, it would be helpful to directly compare the data presented in table 1 and table 2 with prior literature keeping in mind both the sintering temperature and time at temperature.
  8. Results and discussion: Other than achieving close to 100% density, was there any other reason to perform hot isostatic pressing? If not, why is it included? If yes, how does this compare to values in literature? Does precursor method D affect it? Please comment.
  9. Conclusion: The first sentence of the conclusion states that YIP formation was identified from DSC and phase determination. However, no phase information is shown in the paper, making this conclusion unfounded.
  10. Most of the conclusion section simply repeats the observations in the paper. It seems like Method D seems to be the best of all methods shown for achieving a high density. If that is a major conclusion, then the paper should state so explicitly. There is also no insight derived from this data presented in the paper. There needs to be a discussion of why the different methods show the results they do.

Minor comments:

  1. Page 1, abstract: YIG and YIP abbreviations need to be defined at the first instance they are used in the abstract instead of later in the abstract.
  2. Page1, Paragraph 1, Line 4: Maybe use “improved” instead of “increased”.
  3. Page1, Paragraph 2, Line 1: Please cite appropriate references for the numerous literature reports concerning production of monocrystalline and polycrystalline YIG thin films.
  4. Page1, Paragraph 2, Line 2: Please cite appropriate references discussing the interesting properties of dense sinters of YIG.
  5. Page1: Paragraph 3, Line 2: ‘have been’ instead of ‘were’ is more appropriate.
  6. Page 2, Paragraph 3, Line 1: Where it says 8 and 4% Fe2O3, is it mol% or wt%? Please mention in text.
  7. Page 2, Paragraph 4, Line 2: Awkward sentence construction. Please rephrase.
  8. Many symbols in the paper do not appear correctly. Please correct.
  9. Page 3, Paragraph 5, Line 1: Please comment on why Method D powders were not sieved.
  10. Page 3, Paragraph 7, Line 2: Please also include the atmosphere used for DSC measurements. Was it similar to that used for sintering the cylindrical pellets?
  11. The equations on page 4 need their own individual equation numbers.
  12. Page 4: Please define all the values used in equations eg. molecular weight, density.
  13. Page 4: How do the values calculated from equations on page 4 compare to the shrinkage observed in Figure 2?

Author Response

Please find our response letter as a separate PDF file.

Reviewer 2 Report

Yttrium iron garnet is an important material used for amplifier, circulator, insulator and other devices. In this work, the author has fabricated yttrium iron garnet through four different methods and compared the final density by DSC measurement and SEM observation. Overall, I feel that the current work provides useful information for the engineers in material science and can be used to improve the material quality through carefully adjusting the synthesis parameters. Some questions need to be answered before considering publication.
1. In the introduction part, more details about different synthesis methods and their advantages/disadvantages need to be given. Readers from different fields can understand the current situation of yttrium iron garnet synthesis.
2. In the experiment part, more details and main difference between four methods need to be introduced clearly. 
3. In table 2, the grain sizes from different methods look quite similar to me. It is difficult to tell the difference due to different synthesis methods. The analysis may need to be re-considered.
4. There are some grammar errors and typos in the text. The writing needs to be checked carefully.

Author Response

(The authors gave the same response as above.)

Reviewer 3 Report

Dear Authors, 

The manuscript presents a comparative study of the synthesis of yttrium iron garnets obtained by citrate method. The results are clear and very well commented. My only concern about the manuscript is that the identification of the main phase was performed only by analyzing the DSC curves. Do the authors have the possibility to confirm phase formation by a complementary technique such as X-ray diffraction? Moreover, the elemental composition was not studied. Is there a possibility that the stoichiometry of the obtained ceramic is not the same as the designed one? Please argue. 

I will reconsider my recommendation as soon as I receive answers about my concerns for the manuscript.

Author Response

(The authors gave the same response as above.)

Round 2

Reviewer 1 Report

The author's have addressed all the major issues satisfactorily. With the inclusion of the XRD data and more elaborate discussion now included in the edited manuscript, support the conclusions and improve the scientific soundness.

There are however still a few typographical and/or spelling errors in the manuscript eg.

  • "Grenade" instead of "garnet" in line 189 in the newly added paragraph on page 5.
  • Double period at the end of line 198 on page 5.
  • Different font size in text, especially evident on page 3.

I suggest accepting of the paper after these and any other such minor text editing issues.

Reviewer 3 Report

The manuscript was improved. I recommend publication in present form